# Structural characterization of HypX responsible for CO biosynthesis in the maturation of NiFe-hydrogenase

Norifumi Muraki[1,2,3], Kentaro Ishii[1], Susumu Uchiyama [1,4], Satoru G. Itoh[1,2,3], Hisashi Okumura[1,2,3] & Shigetoshi Aono [1,2,3]*

Several accessory proteins are required for the assembly of the metal centers in hydrogenases. In NiFe-hydrogenases, CO and $CN^-$ are coordinated to the Fe in the NiFe dinuclear cluster of the active center. Though these diatomic ligands are biosynthesized enzymatically, detail mechanisms of their biosynthesis remain unclear. Here, we report the structural characterization of HypX responsible for CO biosynthesis to assemble the active site of NiFe hydrogenase. CoA is constitutionally bound in HypX. Structural characterization of HypX suggests that the formyl-group transfer will take place from $N^{10}$-formyl-THF to CoA to form formyl-CoA in the N-terminal domain of HypX, followed by decarbonylation of formyl-CoA to produce CO in the C-terminal domain though the direct experimental results are not available yet. The conformation of CoA accommodated in the continuous cavity connecting the N- and C-terminal domains will interconvert between the extended and the folded conformations for HypX catalysis.

[1] Department of Creative Research, Exploratory Research Center on Life and Living Systems (ExCELLS), National Institutes of Natural Sciences, 5-1 Higashiyama, Myodaiji-cho, Okazaki 444-8787, Japan. [2] Institute for Molecular Science, National Institutes of Natural Sciences, 5-1 Higashiyama, Myodaiji-cho, Okazaki 444-8787, Japan. [3] Department of Structural Molecular Science, The Graduate University for Advanced Studies, 38 Nishogo-naka, Myodaiji-cho, Okazaki 444-8585, Japan. [4] Department of Biotechnology, Graduate School of Engineering, Osaka University, 2-1 Yamadaoka, Suita, Osaka 565-0871, Japan. *email: aono@ims.ac.jp

Hydrogenases are metalloenzymes that catalyze the oxidation of $H_2$ into electrons and protons and the reduction of protons into $H_2$ reversibly, which are expected as biocatalysts for fuel cells and $H_2$ production for clean and sustainable energy[1]. Based on the differences of metal content and the structure of the active site, they are classified into three groups: FeFe-, NiFe-, and Fe-hydrogenases containing a dinuclear Fe unit linked to a [4Fe-4S] cluster, a hetero dinuclear Ni-Fe cluster, and a mononuclear Fe center, respectively[2–6]. These metal clusters in hydrogenases ligate diatomic ligands, CO for the Fe-hydrogenase, CO and $CN^-$ for FeFe- and NiFe-hydrogenases, as terminal ligands[2–6]. These CO and $CN^-$ are biosynthesized and assembled into the metal clusters, for which several accessory and chaperone proteins are required[2,7–13].

While HydG, which is a member of the radical S-adenosylmethionine (SAM) protein family using a [4Fe-4S] cluster, catalyzes the formation of CO and $CN^-$ using tyrosine as a substrate for the maturation of FeFe-hydrogenases[14,15], the different machinery is utilized for biosynthesis of CO and $CN^-$ for NiFe-hydrogenase. $CN^-$ is biosynthesized by the HypE/HypF complex using carbamoyl phosphate as a substrate[8,16–20]. The $CN^-$ ligands are assembled into the $Fe(CO)(CN)_2$ unit in the HypC/HypD complex functioning as a scaffold[8,21–25]. Though the mechanism of $CN^-$ biosynthesis in the maturation process of NiFe-hydrogenases is elucidated, the molecular mechanism of CO biosynthesis remained elusive.

Bürstel et al. report that two independent pathways for CO ligand synthesis are present for the maturation of NiFe-hydrogenase and that HypX is a key factor for CO biosynthesis in one of the two pathways[26]. They also report that $N^{10}$-formyl-THF is the precursor of CO biosynthesized by HypX based on genetic engineering and isotope labelling experiments, by which the carbon atom of the CO ligand in NiFe-hydrogenase is shown to be originated from the α-carbon of glycine[26]. Though it was clearly shown that HypX is involved in the CO ligand synthesis in their report, the molecular mechanism of CO biosynthesis remained unknown. Here, we determine the crystal structures of HypX with and without a substrate, based on which we elucidate the molecular mechanism of CO biosynthesis by HypX.

## Results

**Structure of HypX.** The recombinant of *Aquifex aeolicus* HypX was a monomer in solution (Supplementary Fig. 1). We obtained diffraction-quality crystals of HypX with two different space groups ($C222_1$ and $P4_12_12$ for Forms I and II, respectively) under crystallization conditions as described in Methods. Their crystal structures were determined at a resolution of 1.80 and 2.40 Å for Forms I and II, respectively (Fig. 1, Supplementary Fig. 2, and Table 1). The Form I and Form II crystals contain two molecules (chains A and B) and one molecule of HypX in their asymmetric units, respectively. We used the structure of the chain A of Form I to discuss HypX structure unless otherwise noticed.

HypX consists of the N-terminal (residues 1–270) and the C-terminal (residues 289–542) domains with the C-terminal tail (residues 543–562) as shown in Fig. 1. The N- and C-terminal domains are linked by a loop (residues 271–288). The N-terminal domain is composed of two subdomains, subdomains A (residues 1–151) and B (residues 182–270), which are linked by a long loop (residues 152–181) (Fig. 2a). The subdomain A consists of six β-strands and five α-helices. It forms a Rossmann-fold with a mixed parallel β-sheet, which is constructed by six β-strands that is sandwiched by two sets of two α-helices. The subdomain B has a barrel-helix framework as an oligonucleotide/oligosaccharide-binding (OB) fold consisting of six β-strands and one α-helix, in which six β-strands form an open barrel-like structure.

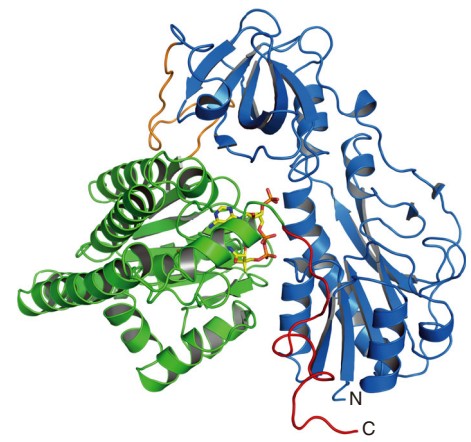

**Fig. 1** Overall structure of HypX (PDB ID: 6J0P). The N-terminal domain (blue) and the C-terminal domain (light green) are linked by a loop (orange). The C-terminal tail is shown in magenta. CoA molecule is shown in the stick model

**Structural homology of the N-terminal domain of HypX.** BLAST[27] search revealed that the amino acid sequence of the N-terminal domain of HypX has a homology to that of the hydrolase domain of $N^{10}$-formyl-tetrahydrofolate ($N^{10}$-formyl-THF) dehydrogenase (FDH)[28]. Protein structure comparison by the DALI server[29] revealed the structural homology between the N-terminal domain of HypX and the hydrolase domain of FDH[30,31], methionyl-t-RNA formyltransferase (FMT)[32,33], and UDP-glucuronic acid dehydrogenase (ArnA)[34–36] (Fig. 2b and Supplementary Fig. 3). The N-terminal domain of HypX showed 19.0–21.0% of amino acid sequence identities to FDH, FMT, and ArnA. They also showed a structural homology as shown in Supplementary Fig. 3. The r.m.s.d. for Cα atoms of the N-terminal domain of HypX was 3.4, 4.9, and 4.5 Å compared with FDH, FMT, and ArnA, respectively.

All of these proteins use $N^{10}$-formyl-THF as a formyl-group donor and catalyze formyl-group transfer reaction[32–38]. Amino acid sequence and structural homologies of the N-terminal domain of HypX to these proteins suggest that it also catalyzes formyl-group transfer reaction with $N^{10}$-formyl-THF as a substrate. If so, HypX will bind THF derivatives in the N-terminal domain. However, neither $N^{10}$-formyl-THF nor THF was bound in the purified HypX. Therefore, we carried out soaking experiments with HypX crystals to confirm whether HypX can bind THF derivatives.

**Binding of THF in the N-terminal domain.** We obtained HypX-THF complex by soaking Form I HypX crystals with THF and solved its structure at a resolution of 2.1 Å as shown in Fig. 3a. The space group ($C222_1$) and the cell dimensions of the crystal were not changed significantly by soaking with THF (Table 1). While the electron density of THF was observed in the chain A, it was not in the chain B. The structure of the chain A is discussed below.

THF is bound at the N-terminal region inside the cavity that is present through the N- and C-terminal domains. The detail of the cavity is described in the section of "Binding of CoA in the C-terminal domain". The interactions between THF and HypX are shown in Fig. 3 and Supplementary Fig. 4. The THF binding sites are conserved among HypX, FDH, and ArnA (Fig. 3b, Supplementary Fig. 3d, e), which supports the notion that $N^{10}$-formyl-THF is the substrate of HypX as is the case of FDH and ArnA.

**Table 1 Data collection and refinement statistics**

**Table 1 (SEPARATED 1/2)**

|  | HypX (Form I) | HypX (Form II) | SeMet-peak (Form II) | THF-bound HypX (Form I) |
|---|---|---|---|---|
| *Data collection* | | | | |
| Space group | $C222_1$ | $P4_12_12$ | $P4_12_12$ | $C222_1$ |
| Cell dimensions | | | | |
| $a, b, c$ (Å) | 79.8, 124.3, 290.9 | 88.3, 88.3, 162.7 | 88.9, 88.9, 163.2 | 80.0 123.7 290.0 |
| $\alpha, \beta, \gamma$ (°) | 90, 90, 90 | 90, 90, 90 | 90, 90, 90 | 90, 90, 90 |
| Wavelength (Å) | 0.90000 | 0.90000 | 0.97910 | 0.90000 |
| Resolution (Å) | 42.47-1.79 (1.86-1.79) [a] | 44.16-2.40 (2.49-2.40) | 42.89-2.40 (2.54-2.40) | 42.31-2.10 (2.18-2.10) |
| Observed reflections | 1,006,078 (98,817)[b] | 373,199 (36,841) | 375,584 (60,954) | 577,575 (55,080) |
| $R_{merge}$[c] | 0.073 (1.094) | 0.096(0.670) | 0.105 (0.656) | 0.071 (0.861) |
| $I/\sigma I$ | 16.77 (1.60) | 19.47 (4.26) | 17.83 (4.04) | 15.24 (1.80) |
| Completeness (%) | 98.8 (89.1) | 99.4 (99.1) | 99.9 (99.7) | 99.88 (99.53) |
| Redundancy | 7.5 (7.5) | 14.5 (14.7) | 14.3 (14.5) | 6.9 (6.7) |
| *Refinement* | | | | |
| Resolution (Å) | 42.47-1.79 | 44.16-2.40 | | 42.31-2.10 |
| Unique reflections | 134,916 | 25,794 | | 84,116 |
| $R_{work}/R_{free}$[d] | 0.173/0.196 | 0.183/0.241 | | 0.178/0.213 |
| No. atoms | | | | |
| Protein | 9306 | 4459 | | 9069 |
| Ligand/ion | 108 | 48 | | 140 |
| Water | 392 | 16 | | 196 |
| *B*-factors | | | | |
| Protein | 35.69 | 56.14 | | 61.37 |
| Ligand/ion | 31.25 | 35.16 | | 58.93 |
| Water | 36.88 | 39.34 | | 49.39 |
| R.m.s. deviations | | | | |
| Bond lengths (Å) | 0.008 | 0.009 | | 0.008 |
| Bond angles (°) | 1.14 | 1.16 | | 0.81 |
| PDB ID | 6J0P | 6J1E | | 6J1F |

**Table 1 (SEPARATED 2/2)**

|  | R9A-Q15A-R131A-R542A | Q15A-R131A-S194A-Q195A-N306A-R542A | A392F-I419F | THF-bound A392F-I419F |
|---|---|---|---|---|
| *Data collection* | | | | |
| Space group | $P4_12_12$ | $C222_1$ | $I222$ | $I222$ |
| Cell dimensions | | | | |
| $a, b, c$ (Å) | 88.5 88.5 161.8 | 79.9 124.4 290.6 | 72.7 139.6 167.7 | 69.4 141.9 170.9 |
| $\alpha, \beta, \gamma$ (°) | 90, 90, 90 | 90, 90, 90 | 90, 90, 90 | 90, 90, 90 |
| Wavelength (Å) | 0.90000 | 0.90000 | 0.90000 | 0.90000 |
| Resolution (Å) | 42.67-2.50 (2.59-2.50) | 47.25-2.10 (2.18-2.10) | 44.85-2.29 (2.38-2.29) | 39.09-2.00 (2.05-2.00) |
| Observed reflections | 314,881 (32,056) | 562,302 (53,424) | 256,066 (23,709) | 2,953,944 (163,839) |
| $R_{merge}$[c] | 0.148 (1.002) | 0.059 (0.842) | 0.045 (0.548) | 0.176 (1.259) |
| $I/\sigma I$ | 12.30 (2.18) | 17.09 (2.12) | 20.13 (2.20) | 14.97 (3.46) |
| Completeness (%) | 99.8 (99.1) | 99.76 (98.51) | 99.47 (95.73) | 99.92 (99.89) |
| Redundancy | 13.7 (14.4) | 6.7 (6.5) | 6.6 (6.5) | 51.5 (39.2) |
| *Refinement* | | | | |
| Resolution (Å) | 42.67-2.50 | 47.25-2.10 | 44.85-2.29 | 39.09-2.00 |
| Unique reflections | 22,960 | 84,408 | 38,512 | 57,352 |
| $R_{work}/R_{free}$[d] | 0.180/0.241 | 0.200/0.237 | 0.176/0.211 | 0.180/0.214 |
| No. atoms | | | | |
| Protein | 4416 | 9152 | 4590 | 4606 |
| Ligand/ion | 54 | 12 | 46 | 92 |
| Water | 12 | 76 | 30 | 80 |
| *B*-factors | | | | |
| Protein | 59.41 | 60.33 | 75.2 | 54.73 |
| Ligand/ion | 63.42 | 65.41 | 111.44 | 78.19 |
| Water | 48.15 | 47.52 | 63.4 | 50.71 |
| R.m.s. deviations | | | | |
| Bond lengths (Å) | 0.009 | 0.009 | 0.008 | 0.008 |
| Bond angles (°) | 1.18 | 0.89 | 1.17 | 1.14 |
| PDB ID | 6J1G | 6J1H | 6J1I | 6J1J |

[a]Values in parentheses are for highest-resolution shell
[b]One crystal was used in each structure except THF-bound A392F-I419F. In THF-bound A392F-I419F structure, a full data set from eight crystals are merged
[c]$R_{merge}(I) = \Sigma I(k) - <I> | / \Sigma I(k)$, where $I(k)$ is the value of the $k$th measurement of the intensity of a reflection, $<I>$ is the mean value of the intensity of that reflection and the summation is the overall measurement
[d]$R_{work} = \Sigma_{hkl}|F_{obs}(hkl) - F_{calc}(hkl)|/\Sigma_{hkl}F_{obs}(hkl)$, where $F_{obs}$ and $F_{calc}$ are the observed and calculated structure factors, respectively. $R_{free}$ is the $R$-factor computed for a test set of reflections that were omitted from the refinement process

There is little structural change in a whole protein upon THF binding. However, slight conformational changes were observed for side chains of Leu56, Lys59, and Tyr108 (Supplementary Fig. 5). THF binding caused relocation of Leu56 and Tyr108

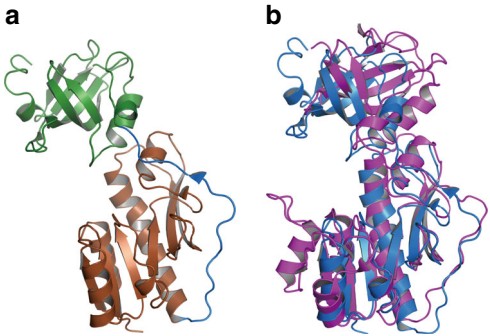

**Fig. 2 a** The structure of the N-terminal domain of HypX, in which the subdomains A (brown) and B (green) are linked by a loop (blue). **b** The superposition of the N-terminal domain of HypX (blue) and the hydrolase domain of FDH (purple, PDB ID: 4ts4)

residues because steric hindrance may occur with THF at the original positions of these residues in THF-free HypX upon THF binding. Lys59 residue shifted toward the domains interface because relocation of Tyr108 caused steric hindrance between Tyr108 and Lys59. The β3-α3 loop in which Leu56 and Lys59 are located also shifted slightly upon THF binding.

**Structural homology of the C-terminal domain.** The C-terminal domain of HypX is composed by a core region of four ββα-motifs consisting eight β-strands and four α-helices, in which CoA is bound as described in the next section (Fig. 4). This core region was surrounded with six helices. BLAST[27] search revealed that the C-terminal domain of HypX shows ~20% of amino acid sequence identity to enoyl-CoA hydratase. Structural homology was detected by DALI server[29] among the C-terminal domain of HypX, enoyl-CoA hydratase (ECH)[39,40], and Δ3-Δ2-enoyl-CoA isomerase (ECI)[41,42] (Supplementary Fig. 6). The r.m.s.d. values compared with the C-terminal domain of HypX are 2.24 Å for the both of ECH and ECI. Enoyl-CoA hydratase and Δ3-Δ2-enoyl-CoA isomerase use CoA/CoA derivatives as their substrates, suggesting that the C-terminal domain of HypX is also a CoA-dependent enzyme. In fact, the C-terminal domain of HypX was found to bind a CoA as described below.

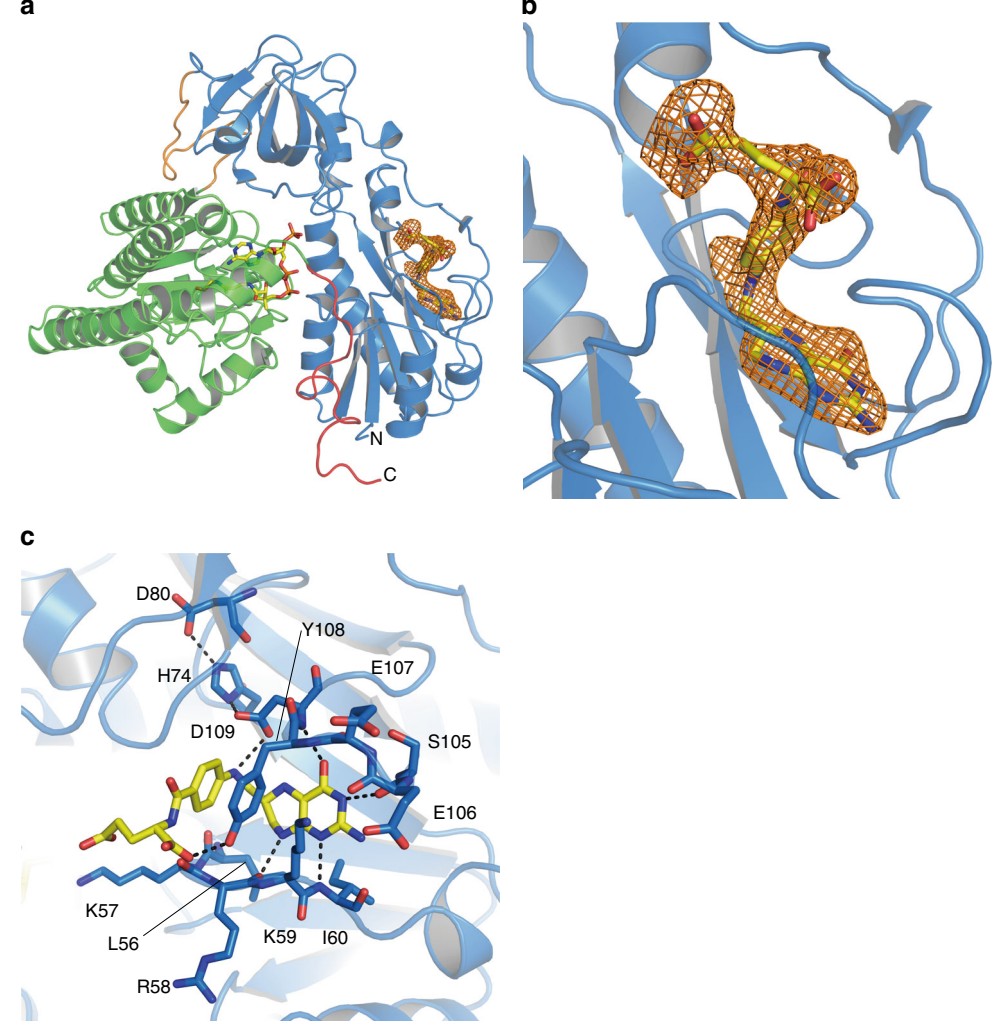

**Fig. 3** Structure of THF-bound HypX (PDB ID: 6J1F). **a** Electron density map (Fo-Fc polder omit map) contoured at 3σ for THF in wild-type HypX, which is shown with an orange mesh. **b** Close-up view of the THF binding region. **c** Interactions between THF and HypX. Pterin ring of THF is sandwiched by the β3-α3 (residues 53–62) and the β5–β6 (residues 103–114) loops. His74, Asp80, and Asp109 form the hydrogen bonding network to fix the orientation of Asp109. Hydrogen bonds are shown in dashed lines

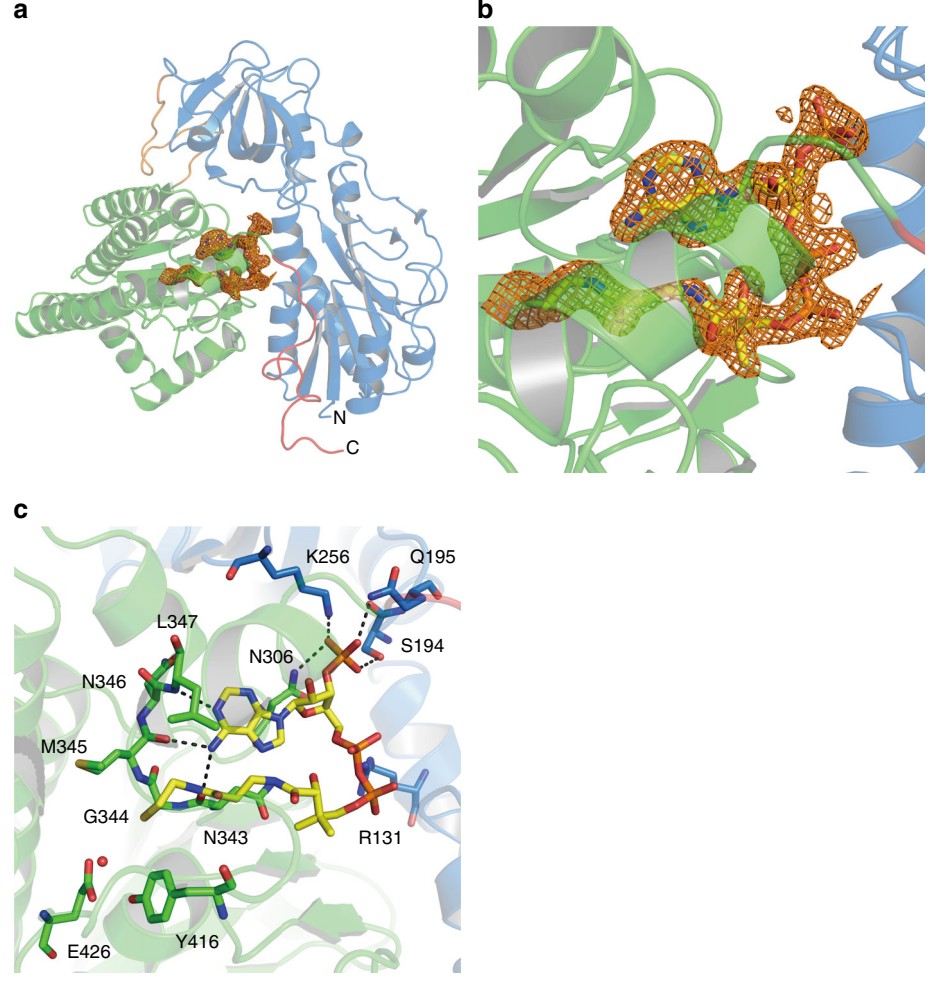

**Fig. 4 a** Electron density map (Fo-Fc polder omit map) contoured at 3σ for CoA in wild-type HypX, which is shown with an orange mesh. **b** Close-up view of the CoA binding region. **c** Interactions between CoA and HypX. CoA and several amino acid residues are shown in the stick model. A red ball stands for the oxygen atom of a water molecule. CoA interacts with amino acids in the both of the N-terminal domain (blue) and the C-terminal domain (light green). Hydrogen bonds are shown in dashed lines

**Binding of CoA in the C-terminal domain**. The N- and C-terminal domains of HypX are linked by a loop (residues 269–287). There are 15 of direct hydrogen bonds and 17 of water-mediated hydrogen bonds in the interface of these two domains (Supplementary Fig. 7 and Supplementary Table 1). A continuous cavity connecting the N- and C-terminal domains is present in the interior of HypX (Fig. 5a). This cavity is open to solvent with two "windows" on the protein surface, one of which is located in the N-terminal domain (window A) and the other at the domains interface (window B), as shown in Fig. 5b.

The unidentified electron density was observed inside the cavity both in the Forms I and II crystals, indicating some molecule is present there. To identify the molecule bound in the cavity, we performed MS analyses of purified HypX. The full $m/z$ range native mass spectra are shown in Supplementary Fig. 8. Native MS revealed that the native and the acid-denatured samples show the molecular mass of 68,299 and 67,531 Da, respectively (Fig. 6a, b). The molecular mass of the native HypX was larger than the denatured one by 767 Da. Consistently, a MS peak of 767 Da was observed for the denatured sample in a negative ion mode MS analysis (Fig. 6c). These results indicate that the molecule of 767 Da is non-covalently bound to HypX presumably in the cavity of HypX.

Based on the results of MS analyses and the shape of the electron density, we assumed the molecule bound in the cavity to

be CoA, whose molecular weight is 767, and found that it was the case. The occupancy and B-factor of CoA were 1.0 and 27.0 Å$^2$, respectively. As we did not add CoA during the purification and crystallization of HypX, it was thought to be bound to HypX in *E. coli* cells. Only CoA-bound HypX was observed in native MS spectra (Fig. 6a), indicating that HypX binds CoA constitutively with a high binding affinity.

CoA is bound in the C-terminal region of the cavity with a "folded conformation" in which adenine and pantetheine groups are stacked in parallel as shown in Fig. 4. There are several hydrogen bonds between adenine group and protein main chain atoms as shown in Supplementary Fig. 9.

**A basic patch for tethering CoA in the cavity**. There are many basic amino acid residues around the windows and inside of the cavity, which make a "basic patch" (Fig. 5b and Supplementary Fig. 10), some of which are involved in hydrogen bonding interactions with CoA (Supplementary Fig. 9). We prepared and determined the crystal structures of variants in which mutations were introduced at amino acid residues interacting with CoA to elucidate physiological roles of these residues. The crystal structure of the R9A-Q15A-R131A-R542A variant (PDB ID: 6J1G), in which four residues interacting with 5'-diphosphate group of CoA were mutated, reveals that this variant binds CoA with the folded

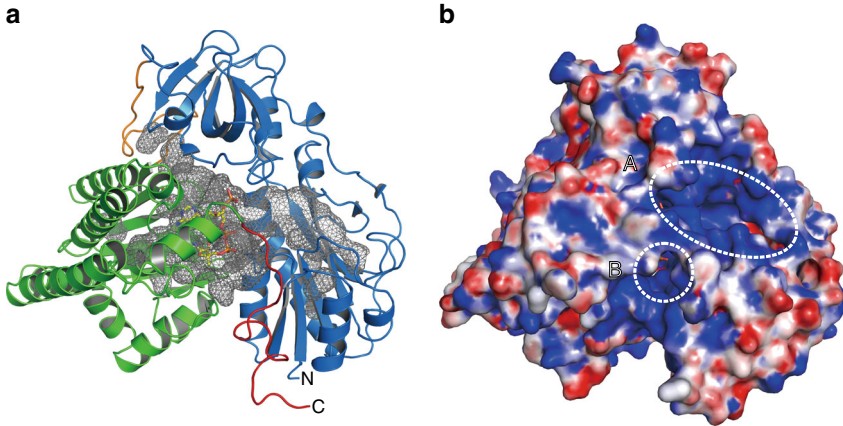

**Fig. 5 a** The continuous cavity connecting the N- and C-terminal domains, which is shown in a mesh. The cavity was represented by interior surface model in PyMol. **b** Surface representation model of HypX with the same orientation as **a**. Two open windows ('A' and 'B' shown in dotted circles) are present on the protein surface

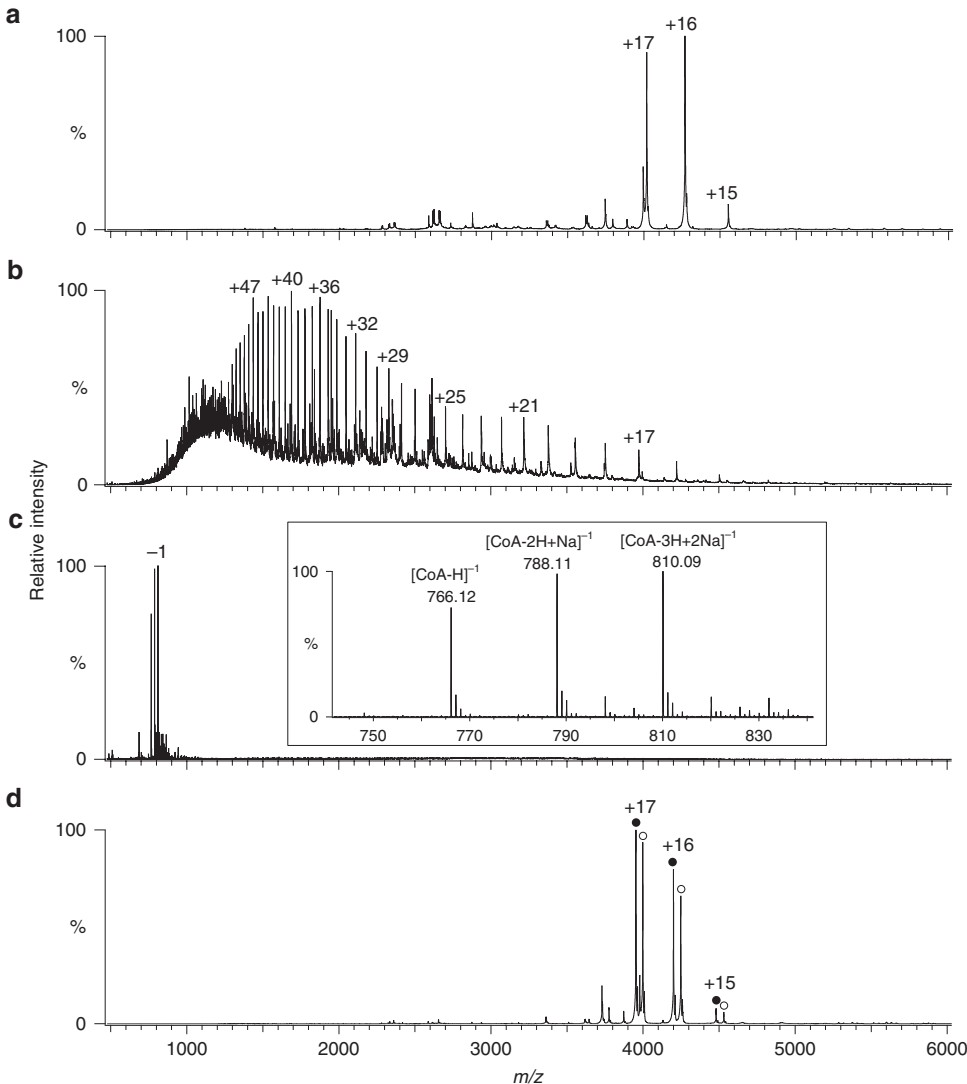

**Fig. 6** Mass spectra of wild-type HypX (**a**, **b** and **c**) and the Q15A-R131A-S194A-Q195A-N306A-R542A variant (**d**), which were measured under **a**, **d** non-denaturing conditions in a positive ion mode, **b** acid-denatured conditions in a positive ion mode, and **c** acid-denatured conditions in a negative ion mode. In the inset of **c**, an enlarged view is shown. CoA-free (closed circles) and CoA-bound (open circles) HypX were observed for the Q15A-R131A-S194A-Q195A-N306A- R542A variant in **d**

conformation as is the case of wild-type HypX (PDB ID: 6J0P and 6J1E), indicating that hydrogen bonding interactions between these residues and 5'-diphosphate group of CoA are not crucial for the recognition and a high binding affinity of CoA by HypX. Further mutations on Ser194, Gln195, and Asn306, which interact with 3'-phosphate group of CoA, resulted in a decrease of CoA binding affinity. The Q15A-R131A-S194A-Q195A-N306A-R542A variant showed two MS peaks due to apo (CoA-free) and holo (CoA-bound) form in native MS analysis while only holo form was observed in wild-type (Fig. 6d). In the crystal structure of this variant (PDB ID: 6J1H), electron density of CoA was not observed probably because CoA was dissociated from the cavity during crystallization. Thus, a loss of hydrogen bonds with 3'-phosphate group of CoA decreases CoA binding affinity of HypX and these hydrogen bonds play an important role for tethering CoA in the cavity.

**Interconversion of CoA conformation**. While CoA adopts the folded conformation in HypX, an "extended conformation" of CoA, in which the ADP and pantetheine moieties are extended in a linear fashion, is observed in some CoA-dependent enzymes[43]. Here we examined whether the extended conformation of CoA was also available in HypX and found that CoA is able to adopt the extended conformation in the A392F-I419F variant. Ala392

and Ile419 are located near the cysteamine moiety of the folded form of CoA, whose positions correspond to "a neck of a bottle" accommodating the pantetheine moiety of CoA in the folded form (Supplementary Fig. 11a). Replacing Ala392 and Ile419 with Phe will narrow "the neck of a bottle" (Supplementary Fig. 11b), which will destabilize the folded conformation of CoA by a steric hindrance.

We solved the crystal structure of the A392F-I419F variant at a resolution of 2.3 Å. Though the electron density of a part of the pantetheine moiety was not observed clearly, that of the ADP moiety was clearly observed, by which we were able to identify the conformation of CoA in this variant (Fig. 7a, b). In this structure, the relative orientation between the adenosine and pantetheine moieties of CoA is different from that in wild-type HypX though the position of the adenosine moiety is the same as wild-type. The pantetheine moiety is extended toward the N-terminal region of the cavity in this variant unlike wild-type HypX.

Reconstruction of hydrogen bonding interactions with the 5'-diphosphate group is observed in the A392F-I419F variant: Arg131 forms a hydrogen bond with O5B of the 5'-diphosphate group in this variant instead of O3A that interacts with R131 in wild-type, and the main chain of Ser12 forms hydrogen bonds with O3A and O4A of the 5'-diphosphate group. These changes in hydrogen bonding interaction will be responsible for stabilization of the extended conformation of CoA.

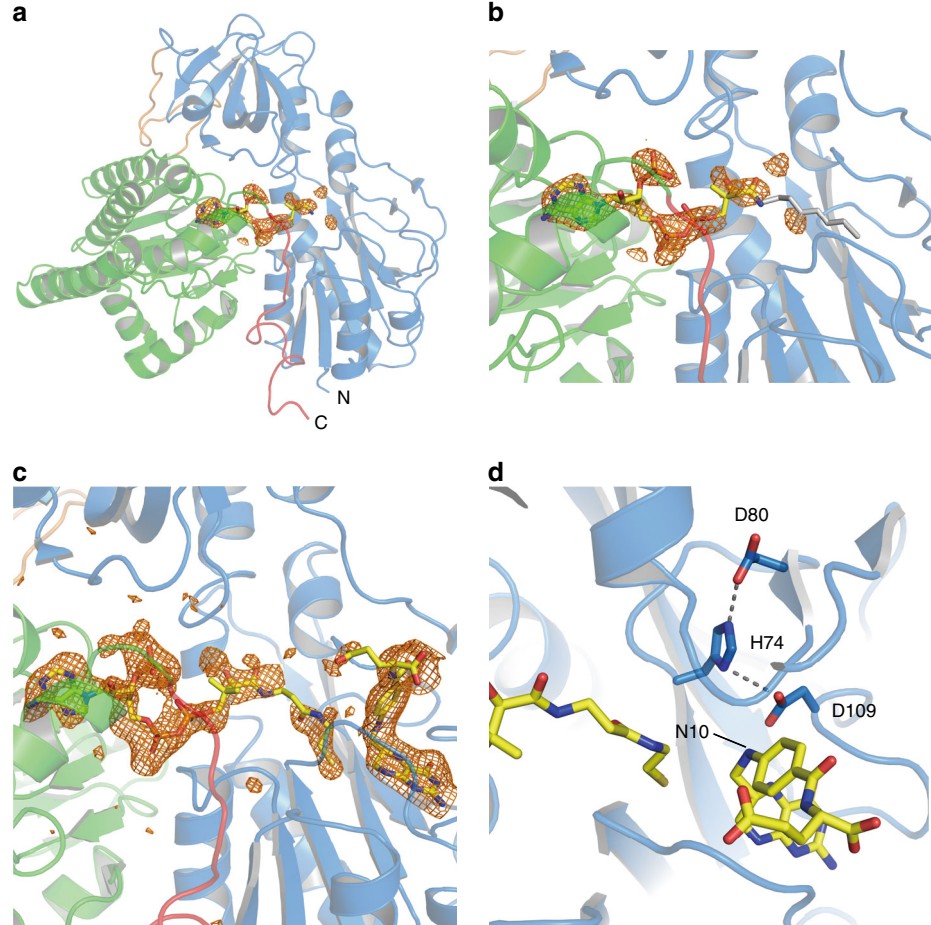

**Fig. 7 a** Structure of the A392F-I419F variant (PDB ID: 6J1I) with the electron density map (Fo-Fc polder omit map shown in an orange mesh) contoured at 3σ for CoA. In the A392F-I419F variant, the occupancy and average B-factor of CoA are 0.89 and 110.4 Å², respectively. **b** Close-up view of CoA binding region in the A392F-I419F variant, in which CoA adopts the extended conformation. **c** Structure of THF-bound A392F-I419F variant (PDB ID: 6J1J). The electron density maps for CoA and THF (Fo-Fc polder omit map shown in orange mesh) are contoured at 3σ. In the THF-bound A392F-I419F variant, the occupancy and average B-factor of CoA are 0.85 and 90.4 Å², respectively. **d** Hydrogen bonding network among the catalytic triad in the A392F-I419F variant. CoA, THF, and side chains of His74, Asp80, and Asp109 are shown in the stick model

We also prepared the A392F-I419F variant binding THF and CoA together by soaking the crystals of this variant with THF and solved the structure at a resolution of 2.0 Å. THF is bound at the N-terminal region of the cavity at the same position as the THF-bound wild-type HypX with the hydrogen bonding interactions identical to those of wild-type. The extended conformation of CoA was retained in this THF-bound variant (Figs. 7c, d). The relocation of Tyr108 that results in inducing the π-π interaction between Tyr108 and THF also takes place in this variant upon THF binding. The β3-α3 loop in the A392F-I419F variant showed little conformational change regardless of the presence or absence of THF unlike wild-type HypX (Supplementary Fig. 12).

**Molecular dynamics simulations**. Though X-ray crystallography showed that CoA in HypX can adopt both of the extended and the folded conformations, dynamic behavior of CoA during its conformational change was unknown. We performed molecular dynamics (MD) simulation to confirm whether CoA in HypX can interconvert between the extended and the folded conformations, and if so, to elucidate its dynamic behavior.

We started the MD simulations using the structure of wild-type HypX, in which CoA adopts the folded conformation. The animation of one of the simulations is shown in Supplementary Movie 1. In Supplementary Fig. 13, the structural analyses at every 20 ns are shown. Little structural change was observed until 60 ns. After 60 ns, the β7-β8 loop (residues 419–423) near the pantetheine moiety shifted toward protein surface by ca. 6 Å, which caused a steric hindrance between the β7-β8 loop and Arg448 to move Arg448 toward protein surface. As Arg448 located in the domains interface interacted with the C-terminal tail, the C-terminal tail simultaneously shifted outward. These changes resulted in an expansion of the cavity around the pantetheine moiety of CoA. Though the β7-β8 loop shifted by 4.6 Å toward the original position after 100 ns, the cavity around the pantetheine moiety remained to be more expanded compared with the starting structure. After 140 ns, deformation of the pantetheine group occurred and the terminal thiol group left the original position. 5'-Diphosphate and 3'-phosphate groups concomitantly shifted toward the N-terminal domain by 2.6 Å and the inside of the cavity by 2.2 Å, respectively. After 160 ns, the β7-β8 loop shifted again toward protein surface with α19 helix, resulting in an expansion of the window B located between the β7-β8 loop and the C-terminal tail. These changes enabled the pantetheine moiety to move to the domains interface from the original position observed in the crystal structure. After 180 ns, CoA adopted a "partially folded" conformation, in which the adenosine and the pantetheine moieties are orthogonal each other (Supplementary Fig. 14). The pantetheine moiety shifted further toward the N-terminal domain compared with the structure after 160 ns and was located in the domains interface. The terminal thiol group was by the window B. After 200 ns, the partially folded conformation of CoA returned to the folded one. Though the conformational changes were observed for the pantetheine moiety, the position and the conformation of the adenosine moiety were rarely different from the original ones throughout the simulation.

The conformational change of CoA from the folded to the partially folded conformations was observed in one of the MD simulations though that from the folded to the extended ones was not achieved, which indicates that CoA in HypX can adopt different conformations dynamically. We remark that CoA maintained the folded conformation in the other two MD simulations. This implies that the time scale of conformational change between the folded to partially folded conformations was longer than 200 ns. To observe the extended form, much longer MD simulation than 200 ns is required.

**Complex formation of HypC, HypD, and HypX**. HypC (10.4 kDa in monomer) and HypD (45.0 kDa in monomer) were eluted from a Superdex75 column with an apparent mass of 21.6 kDa and 35.5 kDa, respectively, indicating that HypC and HypD exist as a homo-dimer and monomer in solution, respectively (Supplementary Fig. 15). The mixture of HypC and HypD was eluted with an apparent mass of 76.4 kDa. Though this result indicates the formation of the complex between HypC and HypD (probably (HypC)$_2$HypD complex), its quaternary structure is not clear at present. The SEC analyses revealed the formation of HypC/HypD and HypC/HypD/HypX complexes (Supplementary Fig. 15). The mixture of HypC, HypD, and HypX was eluted from a Superdex200 column with an apparent mass of 119.7 kDa (Supplementary Fig. 15), suggesting the formation of the 1:1:1 complex of HypC, HypD, and HypX.

**Discussion**

The crystallographic analyses reveal that HypX consists of the N- and C-terminal domains that are structurally homologous to the hydrolase domain of FDH and enoyl-CoA hydratase/isomerase (ECH/ECI), respectively. The hydrolase domain of FDH catalyzes formyl-group transfer from $N^{10}$-formyl-THF to ACP, for which His106, Ser108, and Asp142 act as a catalytic triad[30,31]. At first in this reaction, Asp142 abstracts a proton from the SH group in 4-phosphopantetheine that is a prosthetic group in ACP. The resulting thiolate attacks nucleophilically the carbon atom of formyl group in $N^{10}$-formyl-THF to produce an oxy-anion intermediate. The orientation of Asp142 is fixed by a hydrogen bond with πN of His106[30,31]. Ser106 is responsible for the fixation of the orientation of His106 by a hydrogen bond with τN of His106[30,31]. Formyl group transfer proceeds through this intermediate to produce formylated ACP and THF. The catalytic triad (His, Ser, and Asp) are highly conserved among formyl transferases using $N^{10}$-formyl-THF as a substrate (a donor of formyl group) such as FDH, FMT, and ArnA[30–38].

The comparison of amino acid sequences and crystal structures between HypX and FDH reveals that His and Asp among the catalytic triad are conserved at the corresponding positions (His74 and Asp109 in HypX). Though Ser among the catalytic triad is not conserved in HypX, Asp80 forms a hydrogen bond with τN of His74, which fixes the orientation of His74 as does Ser106 in FDH. Asp80 not only sustain a functional role for the fixation of the orientation of His74 but may also enhance the catalytic activity of Asp109 through the hydrogen bonding network among His74, Asp80, and Asp109. Thus, HypX adopts a slightly modified catalytic triad for formyl-group transfer reaction, indicating that the N-terminal domain of HypX can catalyze formyl-group transfer with $N^{10}$-formyl-THF as a substrate.

While 4-phosphopantetheine in ACP accepts formyl group from $N^{10}$-formyl-THF in the case of FDH[37,38], CoA will do so for HypX because it has the phosphopantetheine moiety identical to ACP. Though we are unable to assign the exact position of the SH group due to the flexibility of the pantetheine moiety of CoA that adopts the extended conformation in the THF-bound HypX, it is probable that the thiol group of CoA can be located near the $N^{10}$ position of THF as shown in Fig. 7d . Based on these results, we propose that formyl-group transfer proceeds from $N^{10}$-formyl-THF to CoA adopting the extended conformation in the N-terminal domain of HypX, for which His74, Asp80, and Asp109 will act as the catalytic triad by the same mechanism as the formyl-group transfer reaction catalyzed by the hydrolase domain of FDH.

Because the $N^{10}$ of THF forms a hydrogen bond with Asp109, the formyl group of $N^{10}$-formyl-THF will lead steric hindrance with Asp109 in the present structure of THF-bound HypX.

**Fig. 8** Reaction scheme of CO formation by HypX. The reaction steps 1–3 take place in the N-terminal domain to form formyl-CoA. The conformational change of formyl-CoA proceeds in the reaction step 4 from the extended to the folded conformations, by which the formyl group will be placed at the active site in the C-terminal domain to form CO. Decarbonylation of formyl-CoA takes place to form CO in the reaction step 5

Comparing the structures between THF-free and THF-bound HypX indicates that conformational changes around the THF binding region are induced by THF binding as shown in Supplementary Fig. 5, which is an induced fit type of conformational change. A similar induced fit type of conformational change will occur upon binding of $N^{10}$-formyl-THF to accommodate the formyl group of $N^{10}$-formyl-THF without any steric hindrance.

These results strongly suggest that formyl-CoA is a reaction intermediate in the CO biosynthesis by HypX. So far, it is only reported that formyl-CoA functions as a substrate of formyl-CoA: oxalate CoA transferase that catalyzes the transfer of a CoA moiety between formyl-CoA and oxalate, by which the formyl group is converted into formate[44–46]. Given that CO is formed from formyl-CoA, there is no precedent for such a reaction. However, it is reported that decarbonylation of phenyl formate and its derivatives takes place in the presence of a weak base to form CO and phenol/phenol derivatives[47]. Though formyl-CoA is a thioester, not an ester, CO and CoA may be formed by decarbonylation of formyl-CoA if a similar reaction takes place. If it is the case, formyl-CoA will be a CO precursor in HypX catalysis.

The C-terminal domain of HypX shows the homologous amino acid sequence and crystal structure to those of ECH/ECI. In these proteins, CoA adopts the folded conformation with the same location near the β8-α4 loop as that in HypX, suggesting that the catalysis in the C-terminal domain of HypX proceeds with the folded conformation of CoA. Based on our results that CoA can adopt both of the extended and the folded conformations in HypX, formyl-CoA formed initially in the N-terminal domain will change its conformation from the extended to the folded ones, and then the catalysis in the C-terminal domain will proceeds.

There are Tyr416 and Glu426 near the SH group of CoA adopting the folded conformation (Fig. 4 and Supplementary Fig. 9), which are candidates for a general base responsible for decarbonylation of formyl-CoA. There is an enough space to accommodate the formyl group of formyl-CoA around the SH group of CoA. *Ralstonia eutropha* mutant strains carrying the

mutation at Tyr439 or Glu449 in HypX, which corresponds to Tyr416 or Glu426 in *A. aeolicus* HypX, respectively, show HypX⁻ phenotype[26], suggesting that both of these two residues are involved in decarbonylation of formyl-CoA. In addition to Tyr416 and Glu426, a water molecule is present near these residues as an alternative candidate for a base. Two mechanisms are conceivable at present: in one mechanism, Tyr416 and/or Glu426 act as a general base(s) to abstract a proton from the formyl group in formyl-CoA, and in the other one, Tyr416 and/or Glu426 abstract a proton from a water to form $OH^-$ that acts as a base for decarbonylation of formyl-CoA. In either case, Tyr416 and Glu426 play an important role for the catalysis in the C-terminal domain of HypX.

Taken together, we propose the reaction scheme of CO biosynthesis by HypX as shown in Fig. 8. HypX will catalyze two consecutive reactions, the formyl-group transfer from $N^{10}$-formyl-THF to CoA and decarbonylation of formyl-CoA, in the N- and C-terminal domains, respectively, to produce CO. The putative catalytic residues, His74, Asp80, and Asp109 in the N-terminal domain and Ty416 and Glu426 in the C-terminal domain, are perfectly conserved among HypX homologs listed in the previous report[26]. CoA is constitutively bound in HypX, which will act as the prosthetic group for CO biosynthesis through the formation and the decarbonylation of formyl-CoA.

In our proposed mechanism, the extended and the folded conformations of CoA are involved in the catalysis in the N- and C-terminal domains, respectively, indicating that interconversion of CoA conformation is required for the HypX function. The conformational change of CoA from the folded to the partially folded conformations is observed in MD simulation though the full conformational change is not. The incomplete conformational change will be caused by a steric hindrance between the pantetheine moiety of CoA and the C-terminal tail. If this steric hindrance is eliminated, the full conformational change of CoA may be possible from the folded to the extended conformations. In the THF-bound HypX, the β3-α3 loop in the N-terminal domain shifts toward the C-terminal tail compared with THF-

free HypX. Further shift of the β3-α3 loop may make the C-terminal tail detach from the domains interface. If binding of $N^{10}$-formyl-THF in the N-terminal domain induces such a shift, the steric hindrance between the pantetheine moiety of CoA and the C-terminal tail will be released and then the conformational change of CoA may be completed from the folded to extended conformations. Thus, binding of $N^{10}$-formyl-THF may couple with the conformational change of CoA to initiate the formyl-group transfer reaction.

A similar conformational change of the pantetheine moiety is observed for ACP[48,49], in which 4-phosphopantetheine is covalently attached to proteins unlike CoA in HypX. The results of crystallographic analyses and MD simulation reveal that the adenosine moiety of CoA is fixed in place and acts as an anchor during the conformational change of CoA.

It would be possible that a protein-protein interaction between HypX and the HypC/HypD complex could promote the conformational change of CoA with a concomitant conformational change of HypX. In fact, the SEC analysis revealed the formation of HypC/HypD/HypX complex. However, further structural characterizations of the HypC/HypD/HypX complex should be required to proof the above hypothesis.

CO formed in the innermost cavity of HypX will go out of the cavity via the window A or B on the protein surface. However, the pathway to these windows is blocked up with the pantetheine moiety of CoA adopting the folded conformation just after completing CO formation. If the conformational change of CoA from the folded to the extended conformations takes place, a pathway to the window B is open while the one to the window A is still blocked up with the diphosphate moiety of CoA. Thus, CO release from HypX, which will proceed via the window B, may be coupled with the conformational change of CoA.

CO produced by HypX is used as a ligand of the iron in the $NiFe(CN)_2(CO)$ center of NiFe hydrogenases. The $Fe(CN)_2(CO)$ unit of the NiFe dinuclear center is assembled in the HypC/HypD complex as a scaffold[21–25]. The binding site of the $Fe(CN)_2(CO)$ unit is proposed to be located at the bottom of a tunnel ca. 20 Å deep inside from the protein surface in the HypC/HypD complex[21–25], to which Fe is initially bound and then $CN^-$ and CO ligands bind to the Fe. If CO produced by HypX is diffused into solvent, it will be inefficient for the assembly of the $Fe(CN)_2(CO)$ unit in the HypC/HypD. It may be a solution to utilize CO produced by HypX effectively is that HypX and HypC/HypD form a complex. If the window B of HypX and the entrance of the tunnel of HypC/HypD are located face to face to form a continuous tunnel between them, produced CO will be channeled without unwanted diffusion from HypX to HypC/HypD. Though the formation of the HypC/HypD/HypX complex revealed by the SEC analysis supports the above hypothesis, the structural characterizations of this complex in detail are required to proof it. The structural characterization of this complex is now in progress.

## Methods

**Protein expression and purification**. The synthetic gene of HypX from *Aquifex aeolicus* (hereinafter, referred to as HypX), whose codon was optimized for expression in *E. coli*, was purchased from FASMAC (Supplementary Fig. 1). The three and six deoxyoligonucleotides of "CAT" and "CTCGAG" were added at the upstream of the start codon and at the downstream of the stop codon, respectively, in the synthesized HypX gene to introduce the NdeI and XhoI restriction sites. The NdeI-XhoI fragment containing *hypX* prepared from the synthetic gene by the digestion with NdeI and XhoI was inserted between the NdeI and XhoI restriction sites in pET-15b(+) to construct an expression vector of HypX (pET-HypX). The expression vectors of the HypX variants were prepared with the QuikChange Site-Directed Mutagenesis Kit (Agilent) or QuikChange Multi Site-Directed Mutagenesis Kit (Agilent). The primers used for mutagenesis are listed in Supplementary Table 2. The expression vector for the Q15A variant (pET-Q15A_HypX) was prepared using the QuikChange Site-Directed Mutagenesis Kit with pET-HypX and a set of the primers for Q15A shown in this table. The expression vector for the

R9A-Q15A-R131A-R542A variant (pET- R9A-Q15A-R131A-R542A_HypX) was prepared using the QuikChange Multi Site-Directed Mutagenesis Kit with the template plasmid (pET-Q15A_HypX) and the primers for R9A, R131A, and R542A shown in Supplementary Table 2.

The template plasmid (pET-Q15A_HypX) and the primers for R131A, S194A-Q195A, N306A, and R542A were used with the QuikChange Multi Site-Directed Mutagenesis Kit for the preparation of the expression vector for the Q15A-R131A-S194A-Q195A-N306A-R542A variant. The expression vector for the A392F-I419F variant was also prepared similarly with the template (pET-HypX) and the primers for N306A and A392F using the QuikChange Multi Site-Directed Mutagenesis Kit.

*E. coli* BL21(DE3) was used as a host for the expression of HypX recombinants. It was grown in 3 ml of LB medium containing ampicillin (100 μg/ml) at 37 °C for 16 h as a preculture. The precultured *E. coli* cells were inoculated into 300 ml of TB medium containing ampicillin to carry out the cultivation at 37 °C for 4 h, and then IPTG (0.5 mM) was added to continue the cultivation at 25 °C for 18 h. *E. coli* cells collected by centrifugation were stored at −85 °C until use.

To purify HypX, the stored *E. coli* cells were resuspended in 50 mM Tris-HCl buffer (pH 8.0) containing 500 mM NaCl, 10% (w/v) glycerol and 100 μM phenylmethylsulfonyl fluoride and were disrupted by sonication. The cell-free extract obtained by centrifugation was loaded on a His-Trap column (GE healthcare), and adsorbed proteins were eluted by imidazole. Fractions containing HypX were collected and subsequently purified by a Hi-TrapQ column (GE healthcare). HypX was finally purified by a Superdex 200 16/60 column (GE healthcare) using 50 mM Tris-HCl buffer (pH 8.0) containing 200 mM NaCl as an eluent. To prepare the selenomethionine (SeMet)-substituted HypX, the methionine-auxotroph *E. coli* BL21-CodonPlus (DE3)-RIL-X was used as a host. The recombinant HypX proteins were expressed with a N-terminal $His_6$ tag.

The recombinants of *A. aeolicus* HypC and *A. aeolicus* HypD were also expressed in *E. coli* BL21(DE3) and purified by Ni-affinity chromatography and size exclusion chromatography (SEC). HypC was prepared as a C-terminal $His_6$-tagged protein while a N-terminal $His_6$-tagged HypD was prepared. The molecular mass of the samples was estimated by the SEC analyses using a Superdex 75 10/300GL (GE healthcare) and a Superdex200 10/300GL column (GE healthcare). These columns were equilibrated with 50 mM Tris-HCl (pH 8.0) containing 200 mM NaCl and 1 mM DTT. The HypC/HypD and HypX/HypC/HypD complexes were prepared by mixing equal molar amount of each components.

**Crystallization**. The purified HypX was concentrated to ~5 mg/mL by a centrifugal filter unit (Amicon Ultra, Merck) to prepare the samples for crystallization experiments. Diffraction-quality crystals were obtained at 20 °C in two different conditions using the sitting-drop method. Plate-shape crystal (Form I) and rod-shape crystal (Form II) were obtained at 20 °C in 12% polyethylene glycol 3350, 0.1 M HEPES (pH7.0), 10% glycerol, and 4% polypeptone and in 20% polyethylene glycol 3350, 0.1 M HEPES (pH7.0), and 1% polypeptone, respectively. The crystal of the SeMet-substituted HypX was obtained in 12.7% polyethylene glycol 3350, 0.055 M HEPES (pH7.4), 8% glycerol, and 1.1% polypeptone. The crystal of the R9A-Q15A-R131A-R542A variant was obtained at 20 °C in 15% polyethylene glycol 3350, 0.2 M $KNO_3$, 2% glycerol, and 2% polypeptone. The crystal of the Q15A-R131A-S194A-Q195A-N306A-R542A variant was obtained at 20 °C in 12% polyethylene glycol 3350, 0.1 M HEPES (pH7.0), 5% glycerol, and 2% polypeptone. The crystal of the A392F-I419F variant was obtained at 20 °C in 16% polyethylene glycol 4000, 20% 2-propanol, 8% glycerol, and 4% polypeptone.

**Data collection, structure determination, and analysis**. For data collection under cryogenic conditions, crystals were soaked briefly in a reservoir solution containing 15 % (v/v) glycerol and flash-cooled in liquid nitrogen except for the A392F-I419F variant. The crystals of the A392F-I419F variant were directly flash-cooled in liquid nitrogen. Diffraction data were collected at 100 K on the beamline BL44XU at SPring-8. The data were processed with the XDS program[50]. The initial phases were obtained by single-wavelength anomalous dispersion (SAD) method with selenium using AutoSol[51] implemented in PHENIX[52]. Eleven selenium sites were identified in the asymmetric unit. The initial model was built using Auto-Build[53] and further manual model building was carried out using Coot[54]. The refinement calculations were performed with phenix.refine[55]. After several cycles of refinement and model building, the phase information was transferred to native data set of Form II. Initial phases of the datasets of Form I and all variants were determined by the molecular replacement method using the structure of Form II crystal as the search model using Phaser[56]. The final structure was validated in PHENIX[52]. The Ramachandran plot of the Form I structure showed 97.5% of residues in favored region, 2.4% of residues in allowed region and only one residue in outlier region. The secondary structure assignments were calculated using the program DSSP[57]. The polder maps were calculated using Polder maps program[58] at 3.0 σ for CoA and THF in HypX. The electrostatic potential of HypX was calculated by APBS program[59]. The maximum and minimum kBT/e values are -10 and 10, respectively. Data collection and refinement statistics are shown in Table 1.

**Native mass spectrometry**. Native mass spectrometry (native MS) was performed as described previously[60]. The HypX sample solutions were buffer-exchanged into 500 mM ammonium acetate buffer (pH 6.8) at 4 °C using a centrifugal spin column

(MicroBioSpin-6 column, Bio-Rad, Hercules, California, USA). These buffer-exchanged samples were immediately analyzed by nanoflow electrospray-ionization mass spectrometry using gold-coated glass capillaries made in house (approximately 2–5 μL sample loaded per analysis). The spectra were recorded on a SYNAPT G2-Si HDMS mass spectrometer (Waters, Massachusetts, Milford, USA) in a positive ionization mode at 1.33 kV with a 150 V sampling cone voltage and source offset voltage, 0 V trap and transfer collision energy, and 5 mL/min trap gas flow. To release CoA from HypX protein, formic acid (30% in final) was added to the buffer-exchanged HypX samples. The acid-denatured samples were analyzed in a negative ionization mode with 4 V trap and 2 V transfer collision energy. The other parameters in measurements were same as those of a positive ionization mode analysis. The spectra were calibrated using 1 mg/mL cesium iodide and analyzed using the MassLynx software (Waters).

**Molecular dynamics simulation**. We determined the charges of the CoA atoms by using the restrained electrostatic potential (RESP) fits[61]. Quantum chemical calculations were performed using the Gaussian16 program[62]. Structure optimization and electrostatic potential calculations for CoA were carried out using the B3LYP level with the 6-31 G* basis set. HypX with CoA was put in a cubic unit cell with explicit water molecules to perform molecular dynamics (MD) simulations. We employed the X-ray crystal structure of the wild-type HypX as the initial structure of our MD simulations. The side length of the cubic unit cell was 95.9 Å and periodic boundary conditions were utilized. The volume was fixed in our MD simulations. We used the parameters of the AMBER parm14SB force field[63] for HypX and CoA. The TIP3P rigid-body model[64] was employed for water molecules. The SHAKE algorithm was utilized to constrain bond lengths with the hydrogen atoms of HypX and CoA and to fix the O–H and H–H distances of the water molecules during the simulations. The cutoff distance for the Lennard-Jones potential energy was 12.0 Å. The electrostatic potential energy was calculated by the particle mesh Ewald method[65]. Temperature was controlled at 350 K by the Nosé-Hoover thermostat[66–69]. Because A. aeolicus is one of the most extreme thermophilic bacteria and can grow at 360 K[70], we employed this high temperature for the MD simulations. The multiple-time-step method was employed, and the time steps were taken to be 4.0 fs for interactions between the water molecules and 1.0 fs for other interactions. Three different initial velocities were employed. For each initial condition, an MD simulation was performed for 200.0 ns including an equilibration run for 10.0 ns. The production run was conducted for 570.0 ns ( = 190.0 ns × 3) in total.

**Reporting summary**. Further information on research design is available in the Nature Research Reporting Summary linked to this article.

## Data availability

The datasets generated during and/or analysed during the current study are available from the corresponding author on reasonable request. Full m/z range mass spectra of the native mass spectrometry data shown in Fig. 6 are presented in Supplementary Fig 8.

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

## Acknowledgements

We thank Nicolas Moghaddam, who was an international internship trainee from Chimie Paris Tech in Institute for Molecular Science, for his effort on screening of purification and crystallization conditions for HypX at the preliminary stage of this work. This work was performed using a synchrotron beamline BL44XU at SPring-8 under the Collaborative Research Program of Institute for Protein Research, Osaka University. Diffraction data were collected at the Osaka University beamline BL44XU at SPring-8 (Harima, Japan) (Proposal No. 2017A6757, 2017B6757, 2018A6853, and 2018B6853). We thank Dr. Eiki Yamashita, Dr. Akifumi Higashiura, and Dr. Kenji Takagi for their help during data collection. The MD simulations were performed using Research Center for Computational Science, Okazaki Research Facilities, National Institutes of Natural Sciences. This work was supported by a Grant-in-Aid for Scientific Research (B) 17H03093 to S.A and Joint Research by Exploratory Research Center on Life and Living Systems (ExCELLS).

## Author contributions

S.A. designed research. N.M. performed the preparation of samples and X-ray crystallographic analyses. K.I. and S.U. performed MS analysis. S.I. and H.O. performed MD simulations. N.M., K.I., S.U., S.I., H.O., and S.A. analyzed and interpreted data. N.M., K.I., S.U., S.I., H.O., and S.A. wrote the manuscript.

## Competing interests

The authors declare no competing interests.
