## [Peer Review File · Communications Biology]

Reviewers' comments:

Reviewer #1 (Remarks to the Author):

The manuscript by Muraki et al. reports on structural functional studies of the NiFe hydrogenase accessory protein HypX from *A. aeolicus*. Previous work from Oliver Lenz's laboratory has indicated that HypX is involved in producing the CO ligand of the hydrogenase active site from formyl-THF but the actual mechanism remains undetermined. The authors have solved the HypX structure from crystals belonging to two different space groups. One of these crystals showed the enzyme to contain bound CoA whereas the other was used to obtain a HypX-THF complex by soaking. HypX is clearly divided into N- and C-terminal domains that carry significant homology to the hydrolase domain of FDH and to enoyl-CoA hydratase, respectively. The structures of both HydX complexes were solved showing the binding of THF to the N-terminal domain and of CoA to the C-terminal domain. In addition, Muraki et al. have solved the structure of a HydX variant with both substrates bound in close proximity. Because CoA shows either a folded or extended conformation depending on the crystal structure, the authors also performed MD calculations to study the transition between these two conformations.

By combining all their structural data Muraki et al. have produced a rather convincing model for CO synthesis by HydX from formyl-CoA. The crystallographic work looks well done and the resolution obtained from the different crystals seems sufficient to justify their conclusions. There are, however, several points that will have to be dealt with before their manuscript is fit for publication in *Communications Biology*.

- Fig. 3b. The electron density for THF does not look great. This may be due to the way the complex was obtained. Maybe co-crystallization with THF rather than soaking would result in higher occupancy and better density.
- Fig. 3c. The labels in the stereo image are not properly positioned. This makes it difficult to identify the different residues. This has to be fixed. The same applies to Fig. 4d which, I think, also has the stereo pair swapped (I see the back of the helix in the front).
- I find calling the two CoA conformations « open » and « closed » confusing and inaccurate, mainly when cavities are also being discussed. I suggest to name them « folded » and « extended » (and « partially folded ») instead.
- Do the authors know why the MD simulation did not generate the extended form? They imply that this may be due to steric hindrance from the C-terminal tail. Since this effect seems to be abolished by THF binding to the N-terminal domain, why didn't the authors try the MD simulation with the THF-CoA-HydX complex?
- It shouldn't be too difficult to determine whether HypC/HypD forms a complex with HydX. Why didn't the authors check this? It would certainly lend credence to their model.
- Fig. S7 should be moved to the main text as it summarizes the authors' catalytic model for CO synthesis by HydX.

Reviewer #2 (Remarks to the Author):

The paper by Muraki reports for the first time the crystal structures of HypX, a recently identified maturation factor of [NiFe] hydrogenase, involved in CO biosynthesis. The authors found that HypX constitutively binds CoA as a cofactor. They also determined the crystal structure of the HypX-THF complex, revealing detailed binding mode of THF. Based on the structural results, the authors propose a new model of CoA and N10-formyl-THF based CO biosynthesis.

Overall crystallographic study was nicely executed and provides insight into the function of HypX. However, the proposed model is mainly based on an unclear electron density map (even at 2.5 Å) using an uncharacterized mutant. No strong evidence were provided to support the formation of formyl-CoA and its conversion. MD simulation also failed to present the open

conformation of CoA. Therefore, the authors' insistence that formyl-CoA functions as a CO precursor is interesting but seems to be too speculative. The following issues should be addressed before publication.

Major issues

1) The authors described the open conformation of CoA in a FoFc omit map at 2.5 sigma (Fig 5). In general, a FoFc map at 2.5 sigma contains noise peaks and is less accurate for modeling. A simple omit map tends to be affected by artificial bulk solvent density (Liebschner et al. Acta D, 2017). How did the authors calculate the omit map? The polder map implemented in PHENIX rather than a simple omit map seems to be suitable to assess the omit map. The occupancy and B-factors of CoA in the open conformation should be described. In addition, the authors did not mention why the A392F-I419F variant was used. The authors should explain how to design the A392F-I419F variant and describe its characteristics.

2) In abstract, the authors describe that the formyl-group transfer takes place from N10-formyl-THF to CoA to form formyl-CoA in the N-terminal domain of HypX, for which His74, Asp80, and Asp109 act as the catalytic triad. The resulting formyl-CoA is converted into CO and CoA by decarbonylation of the formyl group, which is catalyzed by Tyr416 and/or Glu426 in the C-terminal domain of HypX.

However, no structural and biochemical results were provided to prove the transfer of the formyl group from N10-formyl-THF to CoA and its conversion to CO. Further results are required to support the proposal. For example, native-MS analysis of HypX A392F-I419F variant may detect the modification of CoA because this variant will trap the intermediate of the reaction. MD simulation of N-10-formyl-THF bound HypX will provide further insights into the dynamic movement of CoA, as the authors discussed (p21, line570-580). Cocrystallization (not soaking) of HypX WT or A392F-I419F variant with THF or 10-formyl-5,8-dideazafolate (stable analogue) also may reveal larger conformational changes of HypX because the observed conformational changes upon THF binding seems to be restricted by crystal packing.

If it is difficult to add any supporting results, the authors should rewrite title, abstract and discussion.

3) Crystal structure of the HypX-THF complex reveals that Asp109 forms a hydrogen bond with N10 of THF. Will the formyl group of N10-formyl-THF lead steric hindrance with Asp109? On the other hand, is it possible that the C-terminal domain of HypX accommodate the formyl group of possible formyl-CoA. The authors should add further description about these points.

Minor issues

1) p6, line 196. "C" and "P" are in italic.

2) p6, line 205. Please describe r.m.s.d values among the three molecules of HypA

3) Fig.1 Please indicate the N-terminus (N) and C-terminus (C) in Fig 1.

4) p8, line 228, p10 line 291. Rey et al (Rey et al Mol Gen Genet 1996) already reported the domain composition and sequence similarity of HypX family. This paper should be cited.

5) p9, line 272-280. The authors should add supplementary figures to show the conformational changes upon THF binding.

6) p11, line 294. Please provides r.m.s.d values between HypX and ECH/ECI and $\Delta 3$ - $\Delta 2$ -enoly-CoA, respectively.

7) Fig.4C Why is the C-terminal domain of HypX superposed with $\Delta 3$ - $\Delta 2$ -enoly-CoA, not ECH?

8) Fig.4D. The provided stereo view is a cross-eyed style. Please replace it a wall-eyed stereo view.

9) Fig. 5a. Please add explanation how the cavity is calculated and represented. The softwares for preparation of structural figures also should be described.

10) Fig. 5b looks like an electrostatic potential map. If so, please provide the software to calculate it and maximum and minimum kBT/e values.

11) p14 line362-367. Please add supplementary figure for the CoA binding in Form II

- 12) p16- line 423-425. Please add supplementary figures showing conformational changes
- 13) The validation report of HypA WT suggests that large error in the geometry of Arg547. The authors should correct it.

Reviewer #3 (Remarks to the Author):

The authors elucidated the structure of HypX both in complex with CoA, in complex with THF by soaking techniques, as well as some relevant mutants, either apo or holo, according to their affinity toward cofactors. The structural results obtained are significant and allowed them to demonstrate and discuss the structural homologies inferred till now in the literature by sequence alignments only. According to these results, coupled to Dynamics simulation, mutagenesis and mass analysis of protein complexes, as isolated from E. coli cultures, they build mechanistic hypotheses about HypX role in the CO ligand evolution and transfer. However, the reactions proposed are hypothetical and cannot be described as a demonstrated mechanism. The authors should be careful, especially at the beginning, in the title and in the abstract but also throughout the entire paper (example: line 557) in their statements/sentences, because the reactions are far from being demonstrated, given the unique role proposed for the involved cofactors. Indeed, so far, no enzymatic reaction has ever been described to combine N10-formyltetrahydrofolate as a C1 donor and acyl-CoA acting as a substrate.

In general, the work results rich in details and structural novelty and significantly contribute to the characterization of hydrogenases accessory enzymes.

However, some experimental and discussion details should be improved, according to the following list of suggestions (see below).

As a final general comment, the work of Shigetoshi Aono could strongly benefit from additional mutagenesis studies. Indeed, if the key residues proposed for the catalysis were mutated in a model organism and the hydrogenase activity measured to evaluate the efficiency of CO transfer by HypX mutants, they could support their analysis and the mechanism of CO production here proposed. But this could go far beyond the aims of the present work.

I suggest the following adjustments and improvements and supporting experiments:

1. Add a table with primers used for cloning and mutagenesis in the supplementary material and cite it accordingly in the main text (materials and methods).
2. Please show the quality of your purifications/samples by SDS-PAGE of the studied enzymes (should be in the supplementary materials)
3. Mutations introduced should be carefully justified when discussed, especially the usage of A392F-I419F (line 412): explain why this variant, why/how you selected it.
4. When the authors speak about monomeric state of your enzyme, they never propose any experimental data, eventually in the supplementary materials (size exclusion profile, standards). Furthermore, they never discussed the interactions/contact surface size observed in the crystals (for example running PISA or similar software) so the sentences about monomeric state sound arbitrary or not sufficiently supported. Please show the size exclusion results in the supplementary materials.
5. Could you discuss the eventual positioning and space feasible for the formyl group that should be present in the actual reaction, either bound to THF or to CoA later in the transfer and decarboxylation reaction?
6. Could you superpose HypX N-terminal domain and any structure of FDH, FMT or ArnA in complex with THF or its derivatives, if feasible, and/or compare them somehow?
7. Please mention the software and the parameters used to calculate internal cavity shown in the figure (line 317, legend).
8. Refinement statistics as well as validation report are not reported in the crystallographic Table for the Semet crystals, just data collections statistics are included. Apparently, the authors have not deposited the SeMet structure. Please complete the reports accordingly or justify why the

decided not to deposit the structure which allowed them obtaining experimental phases/initial model necessary for any of the following data.

9. Stereo view can be avoided, since obsolete. Interactions panels, such as those produced by Ligplot or similar software can be introduced instead of them, maybe in the supplementary material.

10. Dynamics simulation: the authors should justify the temperature level, being 350 K quite a high value. Pressure should be also explicitly declared. Temperature and pressure parameters constrain should be also described for the three different simulations. Finally, the authors mention that the conformational changes have been observed in one of the three simulations. What happens in the other two? This could be significant or can be explained considering the different parameters applied in the three runs. Dynamics simulation results should be summarized with caution, given the hypothetical nature of the deduced conformational flexibility and limited duration of the simulation (200 ns).

11. Line 585-88: Alternatively, a protein-protein interaction dependent conformational change can be hypothesized. Maybe, the interaction with other Hyp maturases could promote it (at least cannot be excluded).

12. The reaction scheme, even if tentative, could be moved to the main text, to make the discussion easier to be followed, offering a synthetic and immediate representation of the reaction mechanism proposed. Maybe, a simplified version where just the sequential steps are proposed.

13. It would be interesting (not mandatory) if the authors propose some pool down or in vitro assays to investigate the interaction of HypX with other Hyp proteins. Indeed,, as supposed by the same authors in the conclusions of the manuscript such an interaction or a direct exchange should be advocated given the toxicity and lability of CO ligand itself. Analogous studies have been successfully done for other hydrogenases maturases.

14. Some sentences should be written in a more clear and/or simple form. In particular lines 195-200, 234-35, 251, 290, 309-310, 344, 386;

15. Do not repeat Phenix software too many times, just mention once and then cite it.

16. Line 101: E coli cultures are not solutions, take care of the terms used

17. Line 120: use the general device name instead of a commercial name

18. Line 266: correct the amino acid abbreviation

19. Line 272 and following: 0.16 angstrom rmsd has no meaning since it is below the error in coordinates positioning of any structures determined with X-Ray crystallography. Such a difference is analogous to the minimal changes you can observed measuring two crystals of the same protein supposed to be identical... Change the sentence accordingly and underline the local differences. Before, in the paragraph where you described the two different crystal forms obtained, where you say the structures are the same, please put the rmsd to prove it.

20. Line 494: I rather suggest proposing a slightly modified and actual catalytic triad of HypX including the Ser to His replacement, since it is a mutation that could guarantee the catalytic activity proposed.

21. I suggest citing also older pioneer works such as:

Mol Gen Genet (1996) 252:237-248 © Springer-Verlag 1996 Luis Rey • Domingo Fernfindez • Belen Brito Yolanda Hernando • Jose-Manuel Palacios Juan Imperial • Tom/is Ruiz-Argiieso The hydrogenase gene cluster of *Rhizobium leguminosarum* bv. *viciae* contains an additional gene (*hypX*), which encodes a protein with sequence similarity to the N^o-formyltetrahydrofolate-dependent enzyme family and is required for nickel-dependent hydrogenase processing and activity

And

hoxX (*hypX*) is a functional member of the *Alcaligenes eutrophus* *hyp* gene cluster. By Buhrke and Friedrich Archives of Microbiology [01 Nov 1998, 170(6):460-463] Type: Comparative Study, Research Support, Non-U.S. Gov't, Journal Article DOI: 10.1007/s002030050667

Point-by-point Responses to the reviewers' comments.

Reviewer #1 (Remarks to the Author):

- Fig. 3b. The electron density for THF does not look great. This may be due to the way the complex was obtained. Maybe co-crystallization with THF rather than soaking would result in higher occupancy and better density.

Though we have been trying to obtain the crystal of THF-bound HypX by co-crystallization, it is unsuccessful so far. We continue to make an effort for co-crystallization, and the result may be reported in the future.

We were able to assign THF at the occupancy of 1.0 even in the present structure obtained by soaking method. Though the average B-factor of THF, which is 92.0 Å², is slightly high, it is acceptable.

The polder map was calculated to show the clear density map for THF in the revised Fig. 3b instead of a simple omit map shown in the original figure.

- Fig.3c. The labels in the stereo image are not properly positioned. This makes it difficult to identify the different residues. This has to be fixed. The same applies to Fig. 4d which, I think, also has the stereo pair swapped (I see the back of the helix in the front).

According to the comment of reviewer #3, the stereo image is changed to a simple figure and the figures of Ligplot⁺ (Fig. S4 and S8) are added to show the interactions between the ligand and protein in the revised manuscript.

- I find calling the two CoA conformations « open » and « closed » confusing and inaccurate, mainly when cavities are also being discussed. I suggest to name them « folded » and « extended » (and « partially folded ») instead.

Calling the CoA conformations are changed in the revised manuscript as suggested by the reviewer.

- Do the authors know why the MD simulation did not generate the extended form? They imply that this may be due to steric hindrance from the C-terminal tail. Since this effect seems to be abolished by THF binding to the N-terminal domain, why didn't the authors try the MD simulation with the THF-CoA-HydX complex?

In one of the three MD simulations, CoA had the partially-folded conformation although CoA maintained the folded conformation in the other two. This implies that the time scale of conformational change between the folded to partially-folded conformations was longer than 200 ns. To observe an extended form, much longer MD simulation than 200 ns is thought to be required to overcome the steric hindrance from the C-terminal tail.

Though we have carried out the MD simulation with the THF-CoA-HypX complex, no conformational change of CoA was observed in 200 ns. This also suggest that a longer time than 200 ns is required for the full conformational change of CoA.

In the revised manuscript, the description of "We remark that CoA maintained the closed conformation in the other two MD simulations. This implies that the time scale of conformational change between the closed to partially folded conformations was longer than 200 ns. To observe an extended form, much longer MD simulation than 200 ns is required." is added at lines 355-359, page 11.

It may be another possibility that full conformational change of CoA is induced by a kinetic conformational change of HypX induced by formyl-THF binding during its binding process. However, as it

is only a hypothesis at present, further experiments should be required to check if this hypothesis is correct.

- It shouldn't be too difficult to determine whether HypC/HypD forms a complex with HydX. Why didn't the authors check this? It would certainly lend credence to their model.

We have done the additional experiments to confirm that HypC/HypD forms a complex with HypX. Size-exclusion chromatography revealed that HypX formed a complex with HypC/HypD, which is shown in Fig. S14 added in the revised manuscript. The explanation of these results is added at lines 484-491, page 14-15 and at lines 523-526, page 15-16 in the revised manuscript.

- Fig. S7 should be moved to the main text as it summarizes the authors' catalytic model for CO synthesis by HydX.

According to the reviewer's comment, Fig. S8 is moved to the main text as Fig. 8 in the revised manuscript.

Reviewer #2 (Remarks to the Author):

Major issues

1) *The authors described the open conformation of CoA in a FoFc omit map at 2.5 sigma (Fig 5). In general, a FoFc map at 2.5 sigma contains noise peaks and is less accurate for modeling. A simple omit map tends to be affected by artificial bulk solvent density (Liebschner et al. Acta D, 2017). How did the authors calculate the omit map? The polder map implemented in PHENIX rather than a simple omit map seems to be suitable to assess the omit map. The occupancy and B-factors of CoA in the open conformation should be described. In addition, the authors did not mention why the A392F-I419F variant was used. The authors should explain how to design the A392F-I419F variant and describe its characteristics.*

According to the reviewer's suggestion, we have recalculated the polder map by PHENIX at 3.0 sigma to exclude the bulk solvent, which is shown in the revised Fig. 5.

In the A392F-I419F variant, the occupancy and average B-factor of CoA are 0.89 and 110.4 Å², respectively. In the THF-bound A392F-I419F variant, the occupancy and average B-factor of CoA are 0.85 and 90.4 Å², respectively. These descriptions are added in the figure caption of Fig. 7 in the revised manuscript.

Ala392 and Ile419 are located near the C5P (see Fig. S8 for the numbering of the carbon) of the pantetheine moiety of the closed (folded) form of CoA, whose positions correspond to "a neck of a bottle" accommodating the pantetheine moiety of CoA in the closed (folded) form. Replacing Ala392 and Ile419 with Phe will narrow "the neck of a bottle", which will destabilize the closed (folded) conformation of CoA by a steric hindrance. And these replacements will make the open (extended) conformation of CoA more favorable because the residues 392 and 419 are no longer interacting with the pantetheine moiety of CoA in the open (extended) conformation. This explanation is added at lines 283-290, page 9 in the revised manuscript.

2) *In abstract, the authors describe that the formyl-group transfer takes place from N10-formyl-THF to CoA to form formyl-CoA in the N-terminal domain of HypX, for which His74, Asp80, and Asp109 act as the catalytic triad. The resulting formyl-CoA is converted into CO and CoA by decarbonylation of the formyl group, which is catalyzed by Tyr416 and/or Glu426 in the C-terminal domain of HypX.*

However, no structural and biochemical results were provided to prove the transfer of the formyl group from N10-formyl-THF to CoA and its conversion to CO. Further results are required to support the proposal. For example, native-MS analysis of HypX A392F-I419F variant may detect the modification of CoA because this variant will trap the intermediate of the reaction. MD simulation of N-10-formyl-THF bound HypX will provide further insights into the dynamic movement of CoA, as the authors discussed (p21, line570-580). Cocrystallization (not soaking) of HypX WT or A392F-I419F variant with THF or 10-formyl-5,8-dideazafolate (stable analogue) also may reveal larger conformational changes of HypX because the observed conformational changes upon THF binding seems to be restricted by crystal packing.

If it is difficult to add any supporting results, the authors should rewrite title, abstract and discussion.

Though we tried to obtain supporting results in additional experiments according to the reviewer's suggestion, it was difficult to obtain them. Therefore, we rewrote title, abstract and discussion.

The title was changed to "Structural characterization of HypX responsible for CO biosynthesis in the maturation of NiFe-hydrogenase".

The abstract was changed as follows: "Several accessory proteins are required for the assembly of the metal centers in hydrogenases. In NiFe-hydrogenases, CO and CN⁻ are coordinated to the Fe in the NiFe

dinuclear cluster of the active center. Though these diatomic ligands are biosynthesized enzymatically, detail mechanisms of their biosynthesis remain unclear. Here, we report the structural characterization of HypX responsible for CO biosynthesis to assemble the active site of NiFe hydrogenase. CoA is constitutionally bound in HypX. Structural characterization of HypX suggests that the formyl-group transfer will take place from N¹⁰-formyl-THF to CoA to form formyl-CoA in the N-terminal domain of HypX, followed by decarbonylation of formyl-CoA to produce CO in the C-terminal domain though the direct experimental results are not available yet. The conformation of CoA accommodated in the continuous cavity connecting the N- and C-terminal domains will interconvert between the extended and the folded conformations for HypX catalysis.”

3) Crystal structure of the HypX-THF complex reveals that Asp109 forms a hydrogen bond with N10 of THF. Will the formyl group of N10-formyl-THF lead steric hindrance with Asp109? On the other hand, is it possible that the C-terminal domain of HypX accommodate the formyl group of possible formyl-CoA. The authors should add further description about these points.

Because the N¹⁰ of THF forms a hydrogen bond with Asp109, the formyl group of N¹⁰-formyl-THF will lead steric hindrance with Asp109 in the present structure, as pointed out by the reviewer. Comparing the structures between THF-free and THF-bound HypX indicates that conformational changes around the THF binding region are induced by THF binding as shown in Fig. S5, which is an induced fit type of conformational change. A similar induced fit type of conformational change will occur upon binding of N¹⁰-formyl-THF to accommodate the formyl group of N¹⁰-formyl-THF without any steric hindrance. This explanation is added at lines 396 - 403, page 12 in the revised manuscript.

There is an enough space to accommodate the formyl group of formyl-CoA in the C-terminal domain as shown in Figs. 4 and S8. This explanation is added at lines 434 - 435, page 13 in the revised manuscript.

Minor issues

1) p6, line 196. "C" and "P" are in italic.

These are revised according to the reviewer's comment.

2) p6, line 205. Please describe r.m.s.d values among the three molecules of HypX

According to the reviewer's comment, r.m.s.d values are added at lines 102 - 104, page 4 in the revised manuscript. The r.m.s.d. values were 0.462 Å, 0.414 Å, and 0.596 Å between the chains A and B, the chain A and Form II, and the chain B and Form II, respectively.

3) Fig.1 Please indicate the N-terminus (N) and C-terminus (C) in Fig 1.

The N-terminus and C-terminus are indicated in Fig. 1 in the revised manuscript.

4) p8, line 228, p10 line 291. Rey et al (Rey et al Mol Gen Genet 1996) already reported the domain composition and sequence similarity of HypX family. This paper should be cited.

We added this paper as a reference. It is cited as ref. 28 in the revised manuscript.

5) p9, line 272-280. The authors should add supplementary figures to show the conformational changes upon THF binding.

The supplementary figure is added as Fig. S5 in the revised manuscript to show the conformational changes upon THF binding.

6) p11, line 294. Please provides r.m.s.d values between HypX and ECH/ECI and $\Delta 3$ - $\Delta 2$ -enoyl-CoA, respectively.

As ECI is an abbreviation of $\Delta 3$ - $\Delta 2$ -enoyl-CoA, the description of “enoyl-CoA hydratase/isomerase (ECH/ECI), and $\Delta 3$ - $\Delta 2$ -enoyl-CoA isomerase” in the original text was revised as “enoyl-CoA hydratase (ECH) and $\Delta 3$ - $\Delta 2$ -enoyl-CoA isomerase (ECI)” in the revised manuscript at line 187, page 6.

The r.m.s.d. values compared with HypX are 2.24 Å for the both of ECH and ECI. While HypX and ECI (PDB ID: 1sg4) are monomer, ECH forms a homotrimer in which the C-terminal three helices are domain-swapped (PDB ID: 1dub). Therefore, the core region except for the C-terminal three helices in ECH was used to calculate the r.m.s.d. value between HypX and ECH.

These explanations are added at lines 188 - 192, page 6 in the revised manuscript.

7) Fig.4C Why is the C-terminal domain of HypX superposed with $\Delta 3$ - $\Delta 2$ -enoyl-CoA, not ECH?

A supplemental figure (Fig. S6) is added for the superposition of the C-terminal domain of HypX with ECH and $\Delta 3$ - $\Delta 2$ -enoyl-CoA isomerase (ECI) in the revised manuscript.

8) Fig.4D. The provided stereo view is a cross-eyed style. Please replace it a wall-eyed stereo view.

According to the comment of reviewer #3, the stereo image is changed to a simple figure (Fig. 4d) and the figure of LigPlot⁺ (Fig. S8) is added to show the hydrogen bonding interactions between the ligand and protein in the revised manuscript.

9) Fig. 5a. Please add explanation how the cavity is calculated and represented. The softwares for preparation of structural figures also should be described.

The cavity was represented by interior surface model in PyMol. This explanation is added in the figure caption of Fig. 5 in the revised manuscript.

10) Fig. 5b looks like an electrostatic potential map. If so, please provide the software to calculate it and maximum and minimum kBT/e values.

The electrostatic potential was calculated by APBS program. The maximum and minimum kBT/e values are -10 and 10, respectively. These explanations are added at lines 593-596, page 17 in the revised manuscript.

11) p14 line362-367. Please add supplementary figure for the CoA binding in Form II

A supplementary figure for the CoA binding in Form II is added in Fig. S8, according to the reviewer's comment.

12) p16- line 423-425. Please add supplementary figures showing conformational changes

Fig. S10 is added to show that little conformation change occurs regardless of the presence or absence of THF in the A392F-I419F variant unlike wild type HypX.

13) The validation report of HypA WT suggests that large error in the geometry of Arg547. The authors should correct it.

The apparent large error in the geometry of Arg547 is caused by a kind of system error for measuring z-score for double conformations of Arg547 in the validation process by PDB. We contacted a PDBj annotator to receive the following message. They will fix this problem in the near future.

(A message from a PDBj annotator)

We presume that you pointed out that z-score of double conformation for Arg547 is extremely high.

It is a kind of system error for measuring z-score using different pair of conformation caused by truncated conformation A.

We will ask to fix this problem afterwards, but it will take some time to reflect.

Unless fixing the system problem, this issue cannot be resolved.

Reviewer #3 (Remarks to the Author):

I suggest the following adjustments and improvements and supporting experiments:

1. Add a table with primers used for cloning and mutagenesis in the supplementary material and cite it accordingly in the main text (materials and methods).

A Supplementary Table 2 is added to list primers. Its explanation is added at line 537, page 16.

2. Please show the quality of your purifications/samples by SDS-PAGE of the studied enzymes (should be in the supplementary materials)

Fig. S1 is added to show the result of SDS-PAGE in the supplementary materials of the revised manuscript.

3. Mutations introduced should be carefully justified when discussed, especially the usage of A392F-I419F (line 412): explain why this variant, why/how you selected it.

Ala392 and Ile419 are located near the C5P (see Fig. S8 for the numbering of the carbon) of the pantetheine moiety of the closed (folded) form of CoA, whose positions correspond to “a neck of a bottle” accommodating the pantetheine moiety of CoA in the closed (folded) form. Replacing Ala392 and Ile419 with Phe will narrow “the neck of a bottle”, which will destabilize the closed (folded) conformation of CoA by a steric hindrance. And these replacements will make the open (extended) conformation of CoA more favorable because the residues 392 and 419 are no longer interacting with the pantetheine moiety of CoA in the open (extended) conformation. This explanation is added at lines 283 - 290, page 9 in the revised manuscript.

4. When the authors speak about monomeric state of your enzyme, they never propose any experimental data, eventually in the supplementary materials (size exclusion profile, standards). Furthermore, they never discussed the interactions/contact surface size observed in the crystals (for example running PISA or similar software) so the sentences about monomeric state sound arbitrary or not sufficiently supported. Please show the size exclusion results in the supplementary materials.

According to the reviewer's comment, the results of SEC analysis is added in Fig. S1 of the revised manuscript.

5. Could you discuss the eventual positioning and space feasible for the formyl group that should be present in the actual reaction, either bound to THF or to CoA later in the transfer and decarboxylation reaction?

Because the N¹⁰ of THF forms a hydrogen bond with Asp109, the formyl group of N¹⁰-formyl-THF will lead steric hindrance with Asp109 in the present structure, as pointed out by the reviewer. Comparing the structures between THF-free and THF-bound HypX indicates that conformational changes around the THF binding region are induced by THF binding as shown in Fig. S5, which is an induced fit type of conformational change. A similar induced fit type of conformational change will occur upon binding of N¹⁰-formyl-THF to accommodate the formyl group of N¹⁰-formyl-THF without any steric hindrance. This explanation is added at lines 396 - 403, page 12 in the revised manuscript.

There is an enough space to accommodate the formyl group of formyl-CoA in the C-terminal domain as shown in Figs. 4 and S8. This explanation is added at lines 434 - 435, page 13 in the revised manuscript.

6. Could you superpose HypX N-terminal domain and any structure of FDH, FMT or ArnA in complex with THF or its derivatives, if feasible, and/or compare them somehow?

Fig. S3 is added in the revised manuscript to show the superposition between HypX N-terminal domain and THF-bound FDH and ArnA. And its explanations are added at line 128 - 132, page 4 and at lines 166 - 169, page 5.

7. Please mention the software and the parameters used to calculate internal cavity shown in the figure (line 317, legend).

The cavity was represented by interior surface model in PyMol. This explanation is added in the figure caption of Fig. 5 in the revised manuscript.

8. Refinement statistics as well as validation report are not reported in the crystallographic Table for the SeMet crystals, just data collections statistics are included. Apparently, the authors have not deposited the SeMet structure. Please complete the reports accordingly or justify why they decided not to deposit the structure which allowed them obtaining experimental phases/initial model necessary for any of the following data.

The SeMet crystals were used to obtain phases as the SeMet and Form II crystals were almost isomorphous as shown in Table S2. The phase information of SeMet crystal was transferred to native data set of Form II at the same resolution, for which the complete refinement of the structure of the SeMet crystal is not required. In the case of phasing with SeMet-substituted crystals, their structures are not necessarily deposited in PDB. Thus, we decided not to deposit the structure of the SeMet-substituted HypX.

9. Stereo view can be avoided, since obsolete. Interactions panels, such as those produced by Ligplot or similar software can be introduced instead of them, maybe in the supplementary material.

According to the reviewer's comment, the stereo images are changed to simple figures and the figures of LigPlot⁺ are added to show the hydrogen bonding interactions between the ligand and protein in the revised manuscript (Figs. 4, 5, S4, and S8).

10. Dynamics simulation: the authors should justify the temperature level, being 350 K quite a high value. Pressure should be also explicitly declared. Temperature and pressure parameters constrain should be also described for the three different simulations. Finally, the authors mention that the conformational changes have been observed in one of the three simulations. What happens in the other two? This could be significant or can be explained considering the different parameters applied in the three runs. Dynamics simulation results should be summarized with caution, given the hypothetical nature of the deduced conformational flexibility and limited duration of the simulation (200 ns).

Because *Aquifex aeolicus* is one of the most extreme thermophilic bacteria and can grow at 360 K (ref. 70), we employed this high temperature for the MD simulations. This sentence is added at lines 190 - 192, page 6 in the revised manuscript. Reference 70 is added in the revised manuscript.

In our MD simulations, not the pressure but the volume was fixed. This sentence is added at lines 623 - 624, page 18 in the revised manuscript.

Difference among three MD simulations was only initial velocities. Temperature, volume, and initial structure were the same. In one of the three MD simulations, CoA had the semi-open conformation although CoA maintained the closed conformation in the other two. This implies that the time scale of conformational change between the closed and semi-open conformations was longer than 200 ns. We add the corresponding comments at lines 355 - 359, page 11 in the revised manuscript.

11. Line 585-88: Alternatively, a protein-protein interaction dependent conformational change can be hypothesized. Maybe, the interaction with other Hyp maturases could promote it (at least cannot be excluded).

Yes, it would be possible that a protein-protein interaction could promote the conformational change of CoA with a concomitant conformational change of HypX. In fact, size-exclusion chromatography revealed that a HypC/HypD/HypX complex was formed as shown in Fig. S14 in the revised manuscript. However, further structural characterizations of the HypC/HypD/HypX complex are required to proof the above hypothesis. These explanations are added at lines 484 - 491, page 14 - 15 in the revised manuscript.

12. The reaction scheme, even if tentative, could be moved to the main text, to make the discussion easier to be followed, offering a synthetic and immediate representation of the reaction mechanism proposed. Maybe, a simplified version where just the sequential steps are proposed.

According to the reviewer's suggestion, the reaction scheme is moved to the main text as Fig. 8.

13. It would be interesting (not mandatory) if the authors propose some pool down or in vitro assays to investigate the interaction of HypX with other Hyp proteins. Indeed, as supposed by the same authors in the conclusions of the manuscript such an interaction or a direct exchange should be advocated given the toxicity and lability of CO ligand itself. Analogous studies have been successfully done for other hydrogenases maturases.

We have done the additional experiments to confirm that HypC/HypD forms a complex with HypX. Size-exclusion chromatography revealed that HypX formed a complex with HypC/HypD, which is shown in Fig. S14 added in the revised manuscript.

The description of "Though the complex formation among HypC, HypD, and HypX (Fig. S14) supports the above hypothesis, the structural characterizations of this complex in detail are required to proof it. The structural characterization of this complex is now in progress." is added at lines 523 - 526, page 15 - 16 in the revised manuscript.

14. Some sentences should be written in a more clear and/or simple form. In particular lines 195-200, 234-35, 251, 290, 309-310, 344, 386;

According to the reviewer's comment, these sentences are re-written in a simple form in the revised manuscript.

15. Do not repeat Phenix software too many times, just mention once and then cite it.

It is revised according to the reviewer's comment.

16. Line 101: E coli cultures are not solutions, take care of the terms used

"The precultured solution" was changed to "The precultured *E. coli* cells".

17. Line 120: use the general device name instead of a commercial name

A commercial name was changed to the general device name (a centrifugal filter unit).

18. Line 266: correct the amino acid abbreviation

A typo was corrected.

19. Line 272 and following: 0.16 angstrom rmsd has no meaning since it is below the error in coordinates positioning of any structures determined with X-Ray crystallography. Such a difference is analogous to the minimal changes you can observed measuring two crystals of the same protein supposed to be identical... Change the sentence accordingly and underline the local differences. Before, in the paragraph where you described the two different crystal forms obtained, where you say the structures are the same, please put the rmsd to prove it.

According to the reviewer's comment, the sentence of "The r.m.s.d. of C_α over the entire protein was 0.16 Å between THF-free and THF-bound HypX, indicating that there is little structural change in a whole protein upon THF binding." was deleted to change it to "There is little structural change in a whole protein upon THF binding." at line 170, page 6.

Fig. S5 is added to show the local difference in the revised manuscript.

According to the reviewer's comment, the r.m.s.d values are added at lines 102 - 104, page 4 in the revised manuscript. The r.m.s.d. values were 0.462 Å, 0.414 Å, and 0.596 Å between the chains A and B, the chain A and Form II, and the chain B and Form II, respectively.

20. Line 494: I rather suggest proposing a slightly modified and actual catalytic triad of HypX including the Ser to His replacement, since it is a mutation that could guarantee the catalytic activity proposed.

According to the reviewer suggestion, this part was changed to "Asp80 not only sustain a functional role for the fixation of the orientation of His74 but may enhance the catalytic activity of Asp109 through the hydrogen bonding network among His74, Asp80, and Asp109. Thus, HypX adopts a slightly modified catalytic triad for formyl-group transfer reaction, indicating that the N-terminal domain of HypX can catalyze formyl-group transfer with N¹⁰-formyl-THF as a substrate." at lines 379 - 384, page 11 - 12 in the revised manuscript.

21. I suggest citing also older pioneer works such as:

x1 Mol Gen Genet (1996) 252:237-248 © Springer-Verlag 1996 Luis Rey • Domingo Fernfindez • Belen Brito Yolanda Hernando • Jose-Manuel Palacios Juan Imperial • Tom/Is Ruiz-Argüeso The hydrogenase gene cluster of Rhizobium leguminosarum bv. viciae contains an additional gene (hypX), which encodes a protein with sequence similarity to the N~o-formyltetrahydrofolate-dependent enzyme family and is required for nickel-dependent hydrogenase processing and activity

And

x2 hoxX (hypX) is a functional member of the Alcaligenes eutrophus hyp gene cluster. By Buhrke and Friedrich Archives of Microbiology [01 Nov 1998, 170(6):460-463] Type: Comparative Study, Research Support, Non-U.S. Gov't, Journal Article DOI: 10.1007/s002030050667

These works are cited as ref. 13 and 28 in the revised manuscript.

REVIEWERS' COMMENTS:

Reviewer #1 (Remarks to the Author):

The authors have corrected their manuscript as suggested. I have only one remark concerning the stereo figures. I do not agree with the comments of the reviewer that consider this type of figure obsolete. If you see stereo (like 95% of the population) the figure will be much more informative than a flat ligplot. Consequently, I think the authors could have kept their (corrected) stereo images. However, mono images will do.

Reviewer #2 (Remarks to the Author):

The authors have appropriately addressed my concerns.
I would request the authors to make several minor changes.

1. Page 9, "neck of a bottle" is still unclear from LigPlot representation. It will be better that residues Ala392 and Ile419 are also shown in Fig 4c and Fig.7
2. Author should describe calculation of polder maps in the Methods section and legends.
3. SEC analysis of interaction between HypC, HypD and HypX is interesting and impressive. The authors should describe the SEC results in the result section and preparation of HypC and HypD in the Methods section (Are these two proteins also from *Aquifex aeolicus* ?).
4. In Fig. 7c legend, the polder map around THF is shown in orange, not green in the revised 7c.

Reviewer #3 (Remarks to the Author):

The authors carefully revised the paper and most of the major issues have been addressed by the introduced changes. Minor issues were also taken into account. Pictures/schemes were updated and/or shifted accordingly.

The results are well described and carefully discussed.

Best regards

Response to the comment of Reviewer #1.

According to the suggestion of the Editor and Reviewer #1, the stereo images are added as Supplementary Figs. 4b and 9b.

Response to the comments of Reviewer #2.

1. Page 9, "neck of a bottle" is still unclear from LigPlot representation. It will be better that residues Ala392 and Ile419 are also shown in Fig 4c and Fig.7

As I am afraid that adding Ala392 and Ile419 in Figs. 4c and 7 makes these figures too busy, a new figure is added as Supplementary Fig. 11 to show the location of the residues at 392 and 419 in wild type and the A392F-I419F variant.

2. Author should describe calculation of polder maps in the Methods section and legends.

Description for the calculation of polder maps are added in the Methods section (lines 511-512, page 21) and figure legends in Fig. 3, Fig. 4, and Fig. 7.

3. SEC analysis of interaction between HypC, HypD and HypX is interesting and impressive. The authors should describe the SEC results in the result section and preparation of HypC and HypD in the Methods section (Are these two proteins also from Aquifex aeolicus ?).

According to the reviewer's suggestion, explanations on the SEC experiments and results are added in the Result (lines 283-294, page 12) and Methods (lines 468-476, pages 19-20). HypC and HypD are from *A. aeolicus*. This is also described in the revised text.

4. In Fig. 7c legend, the polder map around THF is shown in orange, not green in the revised 7c.

The mistake was corrected.